

# Spatiotemporal analysis of flash flooding events in mountainous area of China during 1950–2015

Nan Wang[1,2], Weiming Cheng[1,2,3,4], Min Zhao[1,4,5], Qiangyi Liu[1,2], Jing Wang[6], Dongcheng Liu[6]

[1] State Key Laboratory of Resources and Environmental Information Systems, Institute of Geographic Sciences and Natural Resources Research, Chinese Academy of Sciences, Beijing, 100101, China
[2] University of Chinese Academy of Sciences, Beijing, 100049, China
[3] Jiangsu Center for Collaborative Innovation in Geographic Information Resource Development and Application, Nanjing, 210023, China
[4] Collaborative Innovation Center of South China Sea Studies, Nanjing, 210093, China
[5] School of Geographic and Oceanographic Sciences, Nanjing University, Nanjing, 210023, China
[6] Research Institute of Exploration and Development Dagang Oil Field, Tianjin, 300280, China

*Correspondence to*: Weiming Cheng (chengwm@lreis.ac.cn)

**Abstract.** Flash flooding is one of the most destructive natural disasters that occur in mountainous areas. Understanding the spatiotemporal characteristics of flash flooding across China is important for enabling better disaster estimation and prevention on the national scale. To bridge the gap in the research of the spatiotemporal characteristics of flash flooding events (FFEs), based on the longest time series of FFEs in China, this paper used Mann-Kendall (MK) test, wavelet analysis, monthly frequency and index of dispersion (D) to detect the temporal variation, temporal periodic and temporal clustering of FFEs in China. The results indicated that: (1) A marked rising in the number of FFEs in China was detected, with a growth rate of 23.62 per year since 1950; (2) On the large scale, the main periodicity characteristics was approximately 12–25a, with three oscillation periods, and tended to be stable since 1980; On the small scale, the 2–8a time scale was prominent, with two oscillation periods, and tended to be stable since 2006; (3) The intra-annual frequency distribution of FFEs can be divided into three types, right-skew, left-skew and symmetry; (4) The inter-annual clustering played the dominant role in FFEs occurrence across China, while the under-dispersions were only detected in six (5%) watersheds. Precipitation anomalies and soil moisture were detected to have a close correlation with FFEs, however, the interplay of climatic variations and anthropogenic activities may impose greatly impacts on the occurrence and evolution of the flash flooding disasters on a large extent. This study provided a preliminary reference for revealing the driving factors of flash flooding disasters in the context of climate change.

## 1 Introduction

Flash flooding is one of the most destructive natural disasters that occur in mountainous areas due to the top ranking of such events among natural disasters in terms of the people affected and the property losses regionally and globally (Borga et al., 2011). Flash flooding disasters are triggered by high-intensity and short-duration rainfall, often of a spatially confined convective origin (Saharia et al., 2017; Smith and Smith, 2015). According to the report by the Ministry of Water Resources




of China (MWRC), up to 72.4% of flood-related deaths are attributed to flash flooding that occurs in mountainous areas (MWRC, 2014). Flash flooding disaster is expected to increase in frequency and severity, through the impacts of global change on climate, severe weather in the form of heavy rainfall and river discharge conditions (Beniston et al., 2011; Kleinen and Petschel-Held, 2007). Consequently, flash flooding mitigation in mountainous areas is an important part of public safety management and social development (Borga et al., 2011).

Despite the significant threats posed by flash flooding, there is a lack of research into the temporal characteristics of flash flooding on the national scale. Most previous studies related to the multivariate frequency analysis of extreme events assumed temporal stationarity. Several recent studies show that flash flooding characteristics exhibit nonstationary behavior due to climate change, urbanization, land-use change, or water resource structures (Liu and Zhang, 2017; Zhang et al., 2018). A obvious upward trend in flood was driven by increasing precipitation and atmospheric circulation in Germany (Petrow and Merz, 2009). The significant downward trends in the annual maximum flooding in southeast and southwest Australia was found to be associated with the El Niño Southern Oscillation (ENSO) (Ishak et al., 2013). By complying a new dataset, which consists of the longest available flow series from across Europe, the spatial and temporal clustering of flood events across the Europe was studied (Mediero et al., 2015). Based on a pan-European database, the patterns of change in flood timing were detected in six regions all over Europe in the past five decades (Blöschl et al., 2017). In China, the changing flood frequency across the Pearl River basin has been evaluated in the period of 1951–2014 (Zhang et al., 2018). However, few studies have been focused on the spatiotemporal changing of flash flooding on the national scale in China.

Temporal clustering of flash flooding may have considerable consequences for flash flooding estimation, flash flooding design and risk management (Merz et al., 2016), what's more, clustering of catastrophic events is an important issue for the insurance industry when modelling the pricing of insurance contracts (Khare et al., 2015). Hence, it is of utmost importance to understand not only if clustering exists but how clustering changes with the time scale. Flash flooding clustering is typically explained by linkages between flash flooding frequency or magnitude and climate. There are well-organized modes of inter-annual, inter-decadal and lower-frequency climate variability (Barnston and Livezey, 1987). Non-parametric tests are particularly suitable for flash flooding series, as hydrological data are usually non-normal distributed and serially correlated (Kundzewicz and Robson, 2004). Moreover, they are more robust to outliers and do not require any assumption related to the distribution (Hamed and Rao, 1998). A better understanding of the flash flooding intra annual clustering and hence the most probable flash flooding generation processes can therefore assist in the identification of homogeneous regions with a dominant flash flooding season (Hall and Blöschl, 2018). Nevertheless, the huge diversity of China brings about the spatial heterogeneity in the spatial and temporal characteristics of flash flooding, which hinders people the better flash flood estimation and forecasts on the national scale.

First-hand information obtained from hazard observations is the basis for any kind of scientific study and assessment (Wirtz et al., 2014). In this study, the spatiotemporal series dataset, which is consist of the longest available records, has been compiled to help address the need for national flash flooding information based on the national investigation held by China Institute of Water Resources and Hydropower Research (IWHR) for the first time (Liu et al., 2018). For the huge diversity of China, the



spatial and temporal characteristics of flash flooding were analyzed within six geomor-regions respectively. Based on the flash flooding events dataset, the temporal trending, temporal period and temporal clustering were analyzed in details, together with the correlation between FFEs and climatic indicators to reveal the potential indicators in the context of climate change. To our best knowledge, seasonal and annual characteristics of flash flooding events over China have been mapped for the first time. In this study, we address this gap in knowledge through our analysis of the flash flooding evolution in the past 60 years in the entire China.

The paper is organized as follows. We first (Section 2) introduced the datasets used in this study. We then (Section 3) described the method used to detect the trending of FFEs in China. This is followed (Section 4) by the analysis of spatiotemporal characteristics of FFEs, including temporal variation, temporal mutation, temporal periodic, and temporal clustering. We finally (Section5) discussed the typical variations and potential impacts of FFEs trending.

## 2 Datasets

### 2.1 Flash flooding events inventory

The flash flooding events (FFEs) in China for the period of 1950–2015 were provided by the National Flash Flooding Disaster Investigation and Evaluation Project. The project was conducted on a national scale in China by data collection, field investigation and information checking. The FFEs records (occurrence time and location) were collected from the multiple sources, including local chronicles and local Bulletin of flood and drought disasters in China, database released by MWRC and official documents issued by local governmental departments. For further obtaining the specific spatial information (longitude and latitude), some field survey work on the historical flash flooding traces have been done at the watershed scale throughout the country. To the best of our knowledge, this inventory is the first dataset focusing on the national flash flooding disasters in China, which consists of the longest available temporal records. In this study, we selected the FFEs with the full date records in the form of year-month-day since 1950, and eventually, more than 40,000 events were used for subsequent research (Figure1).

### 2.2 Geomorphologic regionalization

Geomorphologic regionalization (geomor-region) is foundational within studies on the spatial differentiation of natural environments, which is an important component within regional geomorphology research. Geomorphologic regionalization data were obtained from China's State Key Laboratory of Resources and Environmental Information Systems (LREIS). Accordingly, based on the regional differentiation of essential geomorphologic types and their genesis, the entire country has been divided into six major geomor-regions (Table1, Figure1).

**Table 1** Description of the six geomorphologic regionalization (geomor-regions) in China

| Name | Abbreviation | Description |
|------|--------------|-------------|



| | | |
|---|---|---|
| Eastern Hilly Plains | EP | It located in the northern part of China comprising low terrains and the largest plain areas. Plains and platforms are dominant features of this region, which has well-developed fluvial accumulation landforms. |
| Southeastern Low-middle Mountains | SEM | It located in the southern part of the low terrain topography and is dominated by low elevation hills and low or middle relief mountains, with only 30% of its area occupied by plains and platforms. |
| Northern and Central Middle Mountains and Plains | NCP | It located in the northeastern part of China's middle terrain topography and is characterized by a plateau landform composed of low or middle relief mountains, hills, platforms, and plains. The loess landform is well developed in this region. |
| Northwestern Middle and High Mountains and Basins | NWB | It located in the northwestern part of the middle terrain topography. It is composed of middle to high mountains, with flattened basins interposed between them, and is characterized by an arid desert geomorphology. Mountains with basins are made up of plains, platforms, and hills. |
| Southwestern Subalpine Mountains | SWM | It located in the southern part of the middle terrain topography. Evidencing a typical karst landform, middle or high mountains with middle or high reliefs are widespread with wide valley basins interspersed between them. |
| Tibetan Plateau | TP | It covers China's high terrain topography. It is composed of plains and high mountains at elevations above 4000 m, accounting for three-forths of the area of this region. It is characterized by glacial and periglacial landforms. |



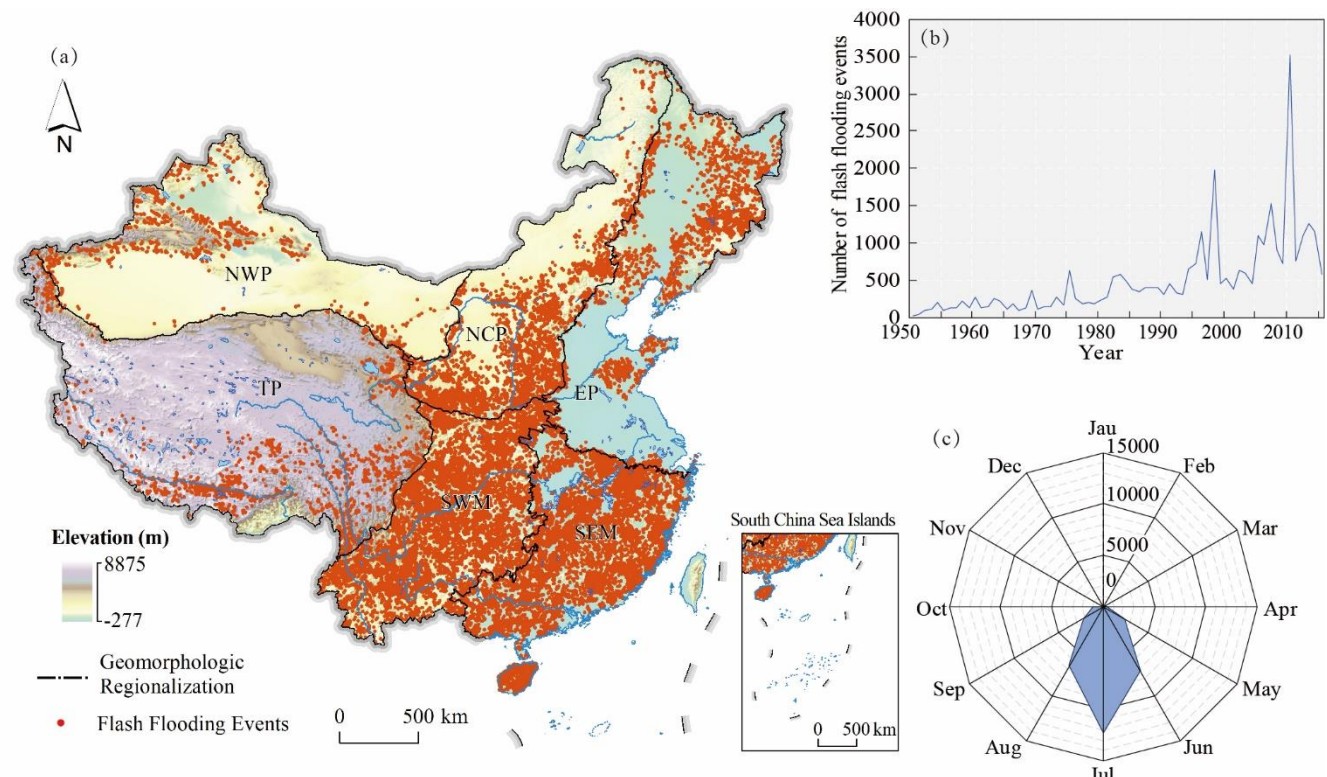

**Figure 1: Location and intra-year and inter-year series of FFEs in China over 1950-2015. (a) the spatial location of the study area and the distribution of FFEs; (b) the intra-year series of the FFEs; (c) the inter-year series of FFEs.**

## 2.3 Watersheds delineation

Watershed was adopted as the basic unit in this study for calculating the temporal mutation and clustering. Based on the third-order stream provided by Resource and Environment Data Cloud Platform (http://www.resdc.cn/), the entire study area was divided and merged into 133 watersheds, with the watershed area ranged from 0.3 to 60 $\times 10^4$ km$^2$.

### 2.4 Climate indicators

The daily precipitation was provided by the China Meteorological Administration (http://data.cma.cn/). In this study, only the stations with the complete data from 1980–2010 were selected for further study. CPC Soil Moisture data provided by the NOAA/OAR/ESRL PSD, Boulder, Colorado, USA, from the Web site at https://www.esrl.noaa.gov/psd/. The monthly data set consists of a file containing monthly averaged soil moisture water height equivalents. The data is model-calculated and not measured directly.



## 3 Methods

### 3.1 Mann-Kendall test and Sen's slope

The Mann-Kendall (MK) test is a nonparametric test method for identifying the increasing or decreasing pattern of the time series, with the advantage of being insensitive to outliers (Kendall, 1948; Mann, 1945). The MK test method has been widely applied to test the significance of the trends in time series (Donat et al., 2013). The equation is as follows:

$$S = \sum_{i=1}^{n-1} \sum_{j=i+1}^{N} sign(X_j - X_i), \tag{1}$$

$$sign(X_j - X_i) = \begin{cases} 1 & X_j - X_i > 0 \\ 0 & X_j - X_i = 0, \\ -1 & X_j - X_i < 0 \end{cases} \tag{2}$$

Where, S is the MK test statistic, $X_i$ and $X_j$ are the values at the time of i and j, respectively; and N is the length of the time series. If a data value from a later time period is higher than a data value from an earlier time period, the statistic S is incremented by 1. On the other hand, if the data value from a later time period is lower than a data value sampled earlier, S is decremented by 1. The final value of S is yielded by the net result of all such increments and decrements (Shahid, 2011). Then, the standard test statistic Z can be computed from Formula to evaluate the presence of a statistically significant trend.

$$Z = \begin{cases} (S-1)/\sqrt{n(n-1)(2n+5)/18} & S > 0 \\ 0 & S = 0, \\ (S-1)/\sqrt{n(n-1)(2n+5)/18} & S < 0 \end{cases} \tag{3}$$

Where, the Z value is greater than 0, indicating that the time series is on rising, vise versa, if the Z value is less than 0, the time series is decreasing; and if the absolute value of the statistic Z is greater than 1.96, this indicates that the trend in a time series meets the 0.05 significance level.

The magnitude of a trend was estimated by the Sen's slope (Sen, 1968). The positive β value represents an increasing trend, while, the negative one means a decreasing trend over the study period. Sen's slope is tested by a two-tailed test at α confidence level.

$$\beta = median\left(\frac{X_j - X_i}{j-i}\right) \forall i < j, \tag{4}$$

### 3.2 Wavelet analysis

Wavelet analysis provides a flexible way to reveal the periodic features of a time series in different time scale by decomposing it into a time-frequency space (Torrence and Compo, 1998). The continuous wavelet transform of a signal f is the convolution of f with a set of scaled and translated wavelets, given as:

$$W_f(a,b) = a^{-1/2} \int_{-\infty}^{\infty} f(t) \psi^* \left(\frac{t-b}{a}\right) dt, \tag{5}$$

Where, $W_f(a,b)$ denotes the wavelet coefficient; ψ is the mother wavelet function, here, Morlet wavelet was chosen as the mother wavelet function; ∗ denotes the complex conjugate; and a, b are the scale and translation parameter.


### 3.3 Monthly frequency

The recorded monthly FFEs frequencies ($FF_m$) can be corrected to account for months with a different number of days (Macdonald and Black, 2010). If no seasonality is assumed, the probability of occurrence of a FFE in a given month is 1/12.

Therefore, in a non-seasonal model $FF_m$ equals 1/12 for any month with upper and lower bounds given by $L_U^N, L_L^N$ for a significance level of 5% (Cunderlik et al., 2004). However, if a monthly frequency is out of these bounds, FFEs do not follow an annual uniform distribution and a seasonal pattern exists at the 5% significance level.

$$FF_m = \frac{F_m}{N}\frac{30}{n_m}, \tag{6}$$

$$L_U^N = \frac{N+11.491}{0.048N^{1.131}}, \tag{7}$$

$$L_L^N = \frac{N-27.832}{0.199N^{0.964}}, \tag{8}$$

Where, $FF_m$ is the frequency of FFEs in the month m, $F_m$ is the number of FFEs recorded in the month m, N is the number of FFEs, $n_m$ is the number of days of month m that, in the case of February, is 28.25 to account for leap years, $L_U^N$ and $L_L^N$ are the upper and lower bounds, respectively, for a non-seasonal population of N FFEs that follows a uniform distribution along the year at a significance level of 5%.

### 150 3.4 Index of dispersion

The occurrence of FFEs can be interpreted as a realization of a point process. A point process which occurs randomly in time is a homogeneous Poisson process, i.e. the event occurrence at any time point is independent of the event occurrences at any previous time point. The degree of event clustering and departure from a homogeneous Poisson process can be characterized by the index of dispersion (D) (Mailier et al., 2006). It relates the variability of FFEs counts to the expectation value of the

counts:

$$D = \frac{Var(Z(T))}{E(Z(T))} - 1, \tag{9}$$

Where, Z(T) is the series of FFEs counts within a time window of length T, Var(Z(T)) is the variance of the FFEs counts and E(Z(T)) is the expected value.

For a homogeneous Poisson, the index of dispersion is equal to zero. Negative values of D stand for under-dispersion and

characterize a more regular pattern of FFEs occurrence than a homogeneous Poisson process. Clustering would be indicated by positive D values which stand for over-dispersion.

### 4 Results

#### 4.1 Temporal variation analysis

The results showed a marked rising in the number of FFEs in China, with a growth rate of 23.62 per year since 1950. The six

geomor-regions showed the similar overall increasing trend, while each geomor-region displayed divergent trending patterns. As the original time series indicated, the number of FFEs reached two obvious peaks at around 1998 and 2010 in all six geomor-regions; EP, SEM and SWM appeared the two significant peaks mentioned above, and showed slight fluctuates in


other periods (Figure 2a, 2b, 2e); however, NCP and NWB showed a growth with more nonstationary changes (Figure2c, 2d); and the number of FFEs in TP increased drastically, especially in the last two decades (Figure2f). As it revealed in the 5-years

and 10-years moving average, intensities in most regions can be divided into two phases, which are 1985–1998 and 2000–2010. NCP and SWM showed a consistent significant increasing trend with the mean change rate at 22.35% and 28.38% per year, respectively; and the mean growth of TP and NWB are 36.6% and 41.26% per year, which showed a steady changing rate; while EP and SEM increased sharply with the mean rate of 79.06% and 68.13% per year. Additionally, the accumulation number for 1950–2015 indicated that the rates of change in FFEs followed three forms: the increasing of EP and NCP showed

the linear pattern; and SEM, NWB and SWM increased at a speeding rate; however, the number of FFEs increased exponentially in TP.

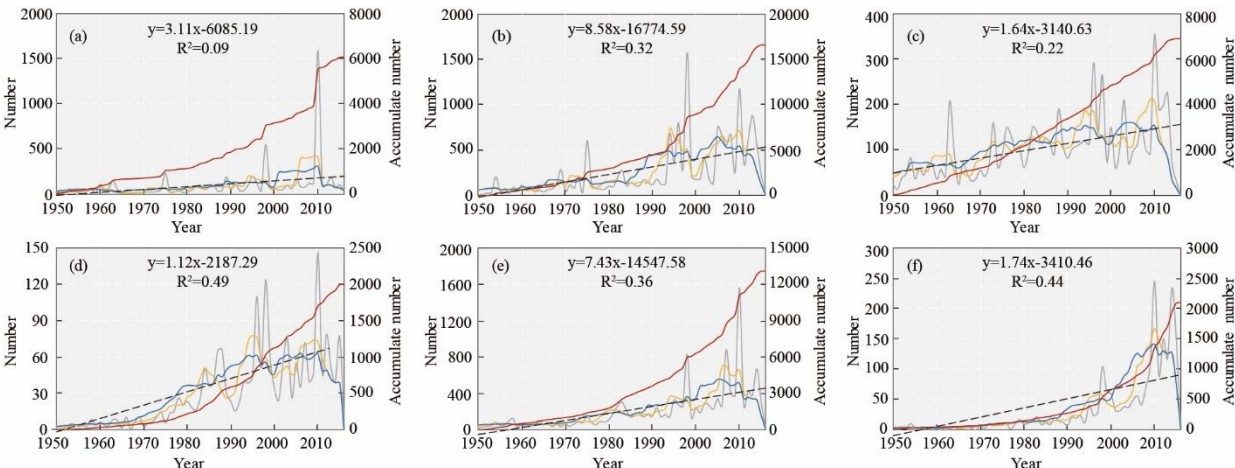

Figure 2: Time series of the annual mean number of FFEs in six geomor-regions of China from 1950 to 2015. The grey line is the original time series; the yellowline and blue line are the 5-year and 10-year moving average of time series, respectively; the red line
is the accumulate number of FFEs; and the black dashed line is the trend line based on the least-squares linear regression. (a) EP; (b) SEM; (c); NCP; (d) NWB; (e) SWM; (f) TP.

**4.2 Temporal mutation analysis**

Trends in the time series of FFEs for the six geomor-regions were tested by MK testing and the magnitude of a trend was estimated by Sen's slope (Figure3). In China, 33% watersheds of all showed the significant upward trends (Table 2). The most
striking upward trends were detected in the two watersheds located in the southeast (Z=7.04) and northeast (Z=7.01) of SWM, while downward trend (Z=-1.28) was only found in the east of EP. Significant upward trends (p<0.1) were detected in all geomor-regions, what stands out in Figure3 was the concentration of significant upward trends in SWM and SEM, with the percentage of 89% and 68% of all watersheds in each geomor-region, respectively. The significant upward trends evenly distributed throughout SWM and located mainly along the southeast coastal area within SEM. In contrast, significant upward
trends were only detected in two watersheds (10%), which located in the northwest of NCP. As the results of Sen's slope estimator test, the median of slope values of EP, SEM, NCP, NWB, SWM, and TP were indicated as 0.37, 2.37, 0.03, 0.23, 2.23, and 0.18, respectively. The statistic results presented an overall agreement between MK testing and Sen's slope estimator in all geomor-regions. The two most obvious increasing was detected in the watershed located in the south and central of SEM, with the Sen's slope value of 0.5. In all, the watersheds with the strong growth trending were mainly distributed in SWM, and
the watersheds showed the strong magnitude but slightly increasing trending were more likely concentrated along the coastal line of SEM.





**Table 2** Number (percentage) of watersheds with upward/downward trends in the number of FFEs.

| Region | EP | SEM | NCP | NWB | SWM | TP | All China |
|--------|-----|------|------|------|------|------|-----------|
| Upward | 4(12%) | 13(68%) | 2(10%) | 5(28%) | 16(89%) | 3(13%) | 44(33%) |
| Downward | 1(3%) | 0(0%) | 0(0%) | 0(0%) | 0(0%) | 0(0%) | 1(1%) |

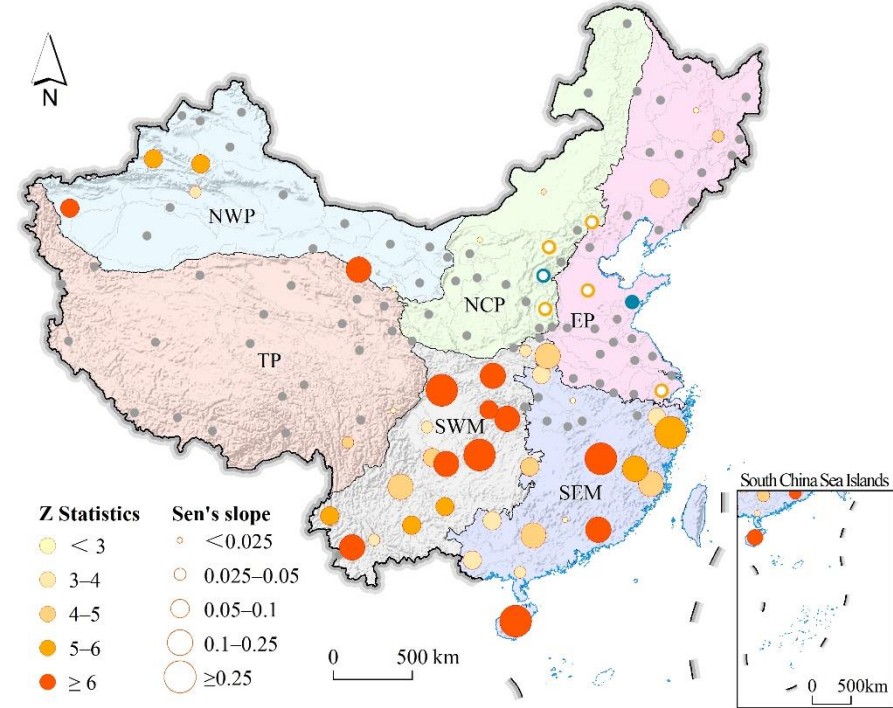

Figure 3: Trends in the annual frequency of FFEs through the MK test and Sen's slope estimation. Where, the blue and yellow solid circles show significant downward and significant upward at a confidence level of 90%; the blue and yellow circles with white holes indicate downward and upward trends that are not satisfied with the confidence interval of 90%; the grey solid circles indicate insignificant trends.

### 4.3 Temporal periodic analysis

### 4.3.1 Period scales identification

The real-part of the wavelet coefficient reflects the periodic variation of the sequence of FFEs. Its distribution in the time domain indicates the future change trending of FFEs on different time scales. The time frequency distribution of the real-part of the wavelet transform coefficients of FFEs in China is shown in Figure4. There are multi-scales variation characteristics in the revolution of FFEs. On the large scale, the main periodicity characteristics is approximately 12–25a, with three oscillation periods, and tend to be stable since 1980. On the small scale, the 2–8a time scale is prominent, with two oscillation periods, which started to stay stable since 2006.

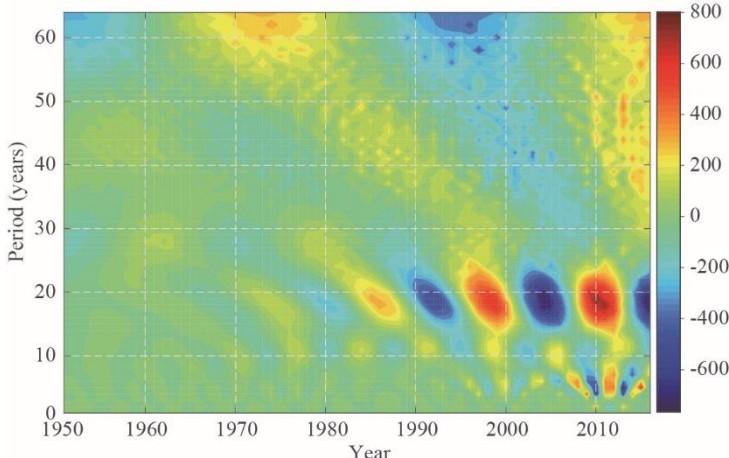

**Figure 4: The real-part of the wavelet coefficient of FFEs.**

The modulus of Morlet wavelet coefficients is the reflection of the distribution of energy density corresponding to the period

of change of different events in the time domain. The larger the coefficient modulus is, the stronger the periodicity of the corresponding time period or scale will be. In Figure 5a, during the evolution of the FFEs, the largest time scale of FFEs is 16–22a, indicating that it is the most obvious temporal scale, followed with the period of the 2–6a, which ranked the second place, and the periodic variation were small. The square of the wavelet coefficients is equivalent to the wavelet energy spectrum, which can analyze the oscillation energy of various patterns of the anomalies. Figure 5b shows that the energy of the 16–22a

time scale is the strongest and the period is the most significant, but its periodic variation has locality, that is, it is mainly concentrated after 2000. The energy of 2–6a time scale took second place, whose main distribution was after 2010.

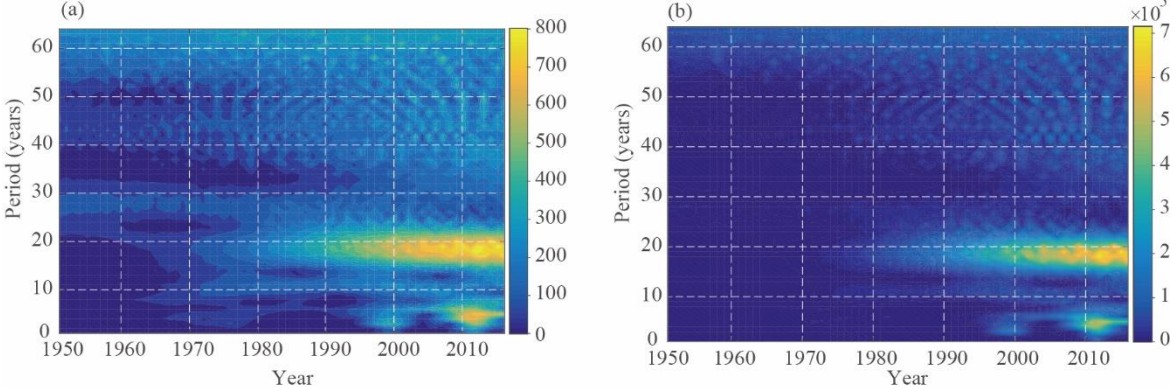

**Figure 5: The Morlet wavelet analysis of FFEs. (a) the modulus of wavelet coefficients; (b) the energy spectrum of wavelet.**

### 4.3.2 Periodic component analysis

The wavelet variance map reflects the distribution of the fluctuation energy of the time series of FFEs with different time scales. It can be used to determine the main period out of the evolution of FFEs. In Figure 6, there were four obvious peaks, which corresponded to the time scales of 19a, 5a, 11a and 3a, respectively. Among them, the most obvious peak corresponded to the 19a time scale, indicating that the period of 19a is the strongest and the first main period of the FFEs variation, followed

by the time scale of 5a, corresponding to the second evolution of FFEs. The fluctuations of the above four cycles controlled
the variation characteristics of FFEs in the overall time domain.

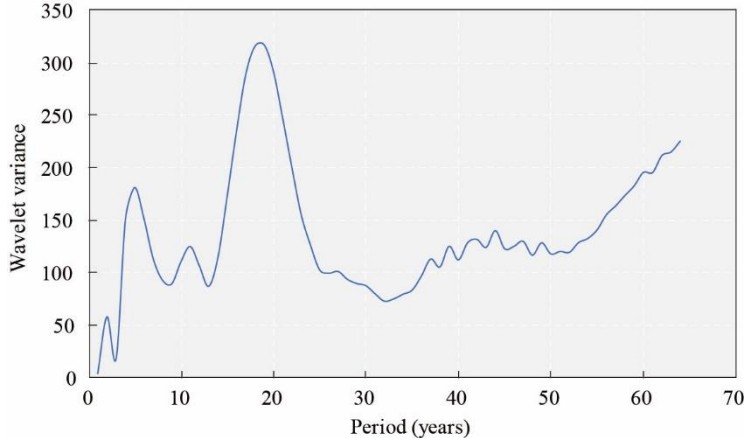

**Figure 6: The wavelet variance analysis of FFEs.**

Based on the test results of wavelet variance, the wavelet coefficients of the first and second main periods that control the
evolution of FFEs was plotted in Figure7. From the main cycle trending, the average period of FFEs, that is, the rich-poor
variation characteristics, can be analyzed under different time scales (Figure 7a). On the characteristic temporal scale of 19a,
the average period of FFEs is about 6 years, and it has experienced about five rich-poor transitional periods. On the
characteristic temporal scale of 5a, the average change period of FFEs is about 2 years, which has experienced a rich-poor
change of 18 cycles (Figure 7b).

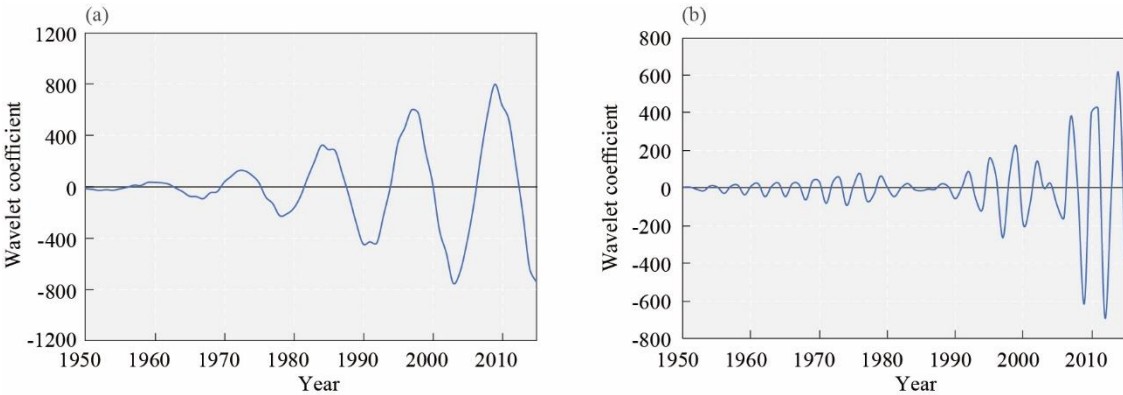

**Figure 7: The wavelet hydrograph in the periodic component of FFEs. (a) on the characteristic temporal scale of 19a; (b) on the characteristics temporal scale of 5a.**

**4.4 Temporal clustering detection**

**4.4.1 Intra-annual clustering**

To assess the FFEs-rich or FFEs-poor months within one year, the seasonality pattern of the monthly frequency of FFEs in six
geomor-regions were analyzed from their mean values in the period 1950–2015. The proposed method tests the significance
of seasons of high and low probability of FFEs occurrences by comparing the observed monthly variability of FFEs




occurrences with the theoretical monthly FFEs variability in a nonseasonal model. Figure8 showed the intra-annual monthly frequency for all watersheds and the areal averaged intra-annual trends for each geomor-region.

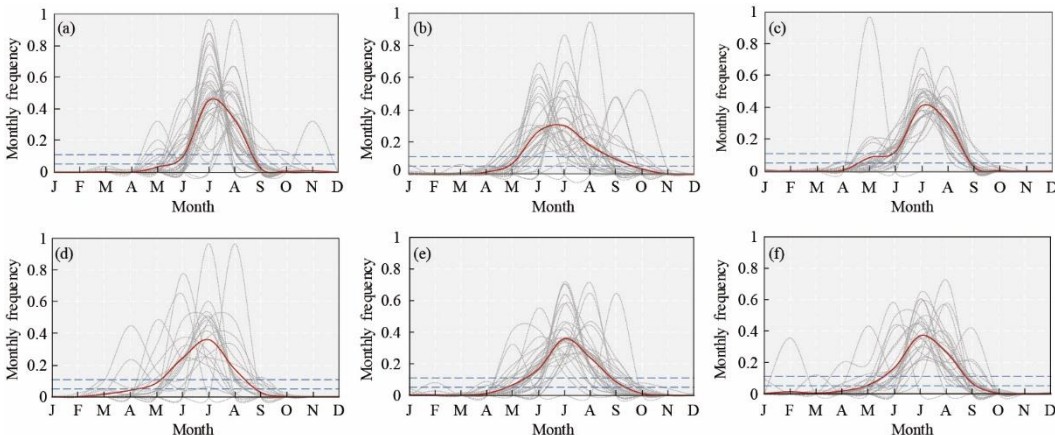

**Figure 8: Monthly frequency of FFEs in six geomor-regions. Where, the grey lines are the monthly frequency of the watersheds within each geomor-region; the red line is the mean monthly frequency of each geomor-region; the horizontal blue dashed line is the confidence intervals of 95%, in the case of a non-seasonality pattern. (a) EP; (b) SEM; (c); NCP; (d) NWB; (e) SWM; (f) TP.**

From the distribution pattern showed in Figure 8 and Table 3, the frequency distribution can be divided into three types, namely, right-skew, left-skew and symmetry. EP, NCP, and TP tend to be right-skew distribution, with the high frequency were more

likely to appear after July. The intensive convective rainfall in these geomor-regions concentrated in mid to late-summer. NCP seemed to lean to left, which meaned the FFEs occurred earlier than the other geomor-regions. The snowmelt caused by the increasing temperature in late-spring may be the proper reason. SEM and SWM were identified as symmetry distribution for the high monthly frequency covering the whole summer.

Table 3 FFEs characteristics of six geomor-regions identified in China.

| Region | Flood-rich months | Flood-poor months | Distribution pattern |
|---|---|---|---|
| EP | July, August | September to May | Right-skew |
| NCP | June to August | September to April | Left-skew |
| NWB | June to August | September to April | Right-skew |
| SEM | June to August | October to April | Symmetry |
| SWM | June to August | October to April | Symmetry |
| TP | June to August | October to April | Right-skew |

Trends for the mean value in six geomor-regions showed the similar variation pattern (Figure 9). In EP, only two FFEs-rich months were identified in July and August, while, a longer FFEs-rich season was observed in the other five geomor-regions, which was from June to August. For the FFEs-poor season, there was an obvious division between the three geomor-regions in the north of China (EP, NCP and NWB), and those in the south of China (SEM, SWM and TP). September to April were detected as the FFEs-poor season in the regions located at high latitudes, when snow accumulates in the watershed and flash

floodings occur less frequently. However, the poor seasons of the regions at low latitudes were observed shorter than above, which started from October to April.



**Figure 9: Maps showing the monthly frequency of FFEs by different months. Where, the color bar represents the monthly FFEs frequency (FFm) for each month; the yellow to red areas are the watersheds with significant FFEs-rich month (FFm larger than 0.11), while the blue areas are the watersheds with significant FFEs-poor month (FFm lower than 0.05).**

### 4.4.2 Intra-annual clustering

To examine the inter-annual FFEs clustering, index of dispersion (D) was applied with the FFEs occurrence frequency data. The D larger than 0 was observed in five out of six geomor-regions across China, indicating that the inter-annual clustering played the dominant role in FFEs occurrence (Figure10). Extensive significant clustering, large D up to 1, can be detected in EP, SEM, and SWM geomor-regions, with the largest one of 3.19 in Hainan Island of SEM. The strength of clustering was fading out from east to west of China. Nevertheless, there were six of all watersheds showing statistically significant results in negative D, which indicated that the inter-annual occurrence of FFEs in these watersheds were under-dispersion. In this study, a more regular distribution of FFEs occurrence was mainly identified in NCP, SEM, and the junction zone between TP and SWM.




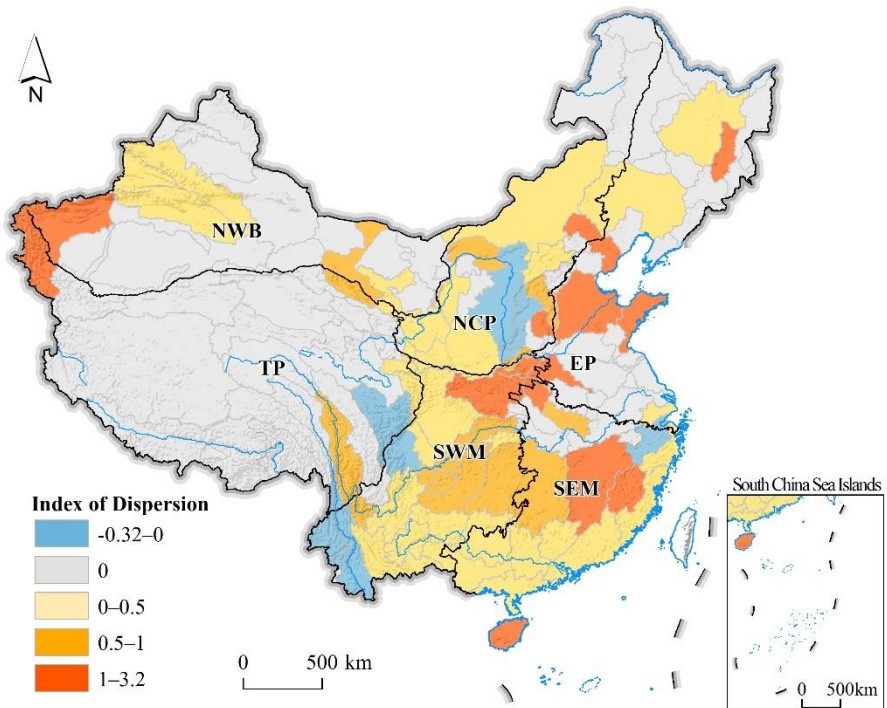

**Figure 10: Index of dispersion (D) of FFEs in watersheds for the period of 1950–2015.**

## 5 Discussion

### 5.1 Potential impacts of the FFEs trending

Although the results in this study showed that the characteristics of FFEs had relatively clear regional patterns due to the
dominance of climatic controls at regional scale, subsurface properties (i.e., catchment storage) also play a considerable role
for the prediction of FFEs, which had not been discussed before. Researchers indicated that climate changing contributed much
to the occurrence and magnitude of the hydro-meteorology hazards in recent years (Liu et al., 2019b; Peng et al., 2019;
Zhang et al., 2014; Zhao et al., 2010). While, how does the climate indicators influence the flash flooding is still unveiled on
the national scale. Among the climate indicators, precipitation and soil moisture are the two prominent factors which may
induce the flash flooding. In this paper, we selected the seasonal precipitation with daily precipitation exceeding 90[th] percentile
of 1980–2010 daily precipitation (R90p) and the mean soil moisture (SM) of summer (May to August) to detect the potential
impacts that may be caused by these indicators.

Figure11 showed the scatter plot of R90p and FFEs in all six geomor-regions, and the significant positive correlations between
R90p and FFEs were identified in SEM (Figure 11b) and NWB (Figure 11d). The variation of atmospheric circulations and
monsoon activities on the large scale have great influence on the regional precipitation. The anomalies of atmospheric
circulations could be the important reasons for the variability of the intensity and frequency of extreme precipitation in China
(Lv et al., 2019; Ma et al., 2018). Studies indicated that the East Asia Summer Monsoon (EASM) was one of the key factors
of the climate system, which severely influenced the precipitation variation in China (Loo et al., 2015; Zhang, 2015).
Additionally, the movement of the rainfall belt was closely connected with the advance and retreat of EASM (Lǖ et al., 2007;





Lu et al., 2013; Qian et al., 2002). SEM and NWB were greatly influenced by the EASM frequently, which are located along the Yangtze River. Usually, EASM appeared in the Yangtze River basin in mid-June, which is called Meiyu period. Meanwhile, the extreme precipitation increased in the mid and lower reaches of the Yangtze River (Huang et al., 2007; Li et al., 2018; Liu et al., 2019a).

Figure 12 displayed the scatter plot of soil moisture and FFEs in all six geomor-regions, and the significant positive correlation 305 between soil moisture and FFEs was identified only in TP (Figure 12f). Soil moisture has an important role in the hydrological cycle, governing the evaporation, runoff, infiltration processes. Therefore, apart from precipitation, the overall soil moisture state of a catchment is another vital factor in the initiation of flash flooding. The snow cover over the TP exhibited a positive trend in winter and spring. Thick snow on the Tibetan plateau in winter increases the soil moisture content in summer and weakens the land-sea thermal contrast over East Asia, which causes the weakening of the subsequent EASM circulation (Tian 310 and Fan, 2013; Zhao et al., 2010).

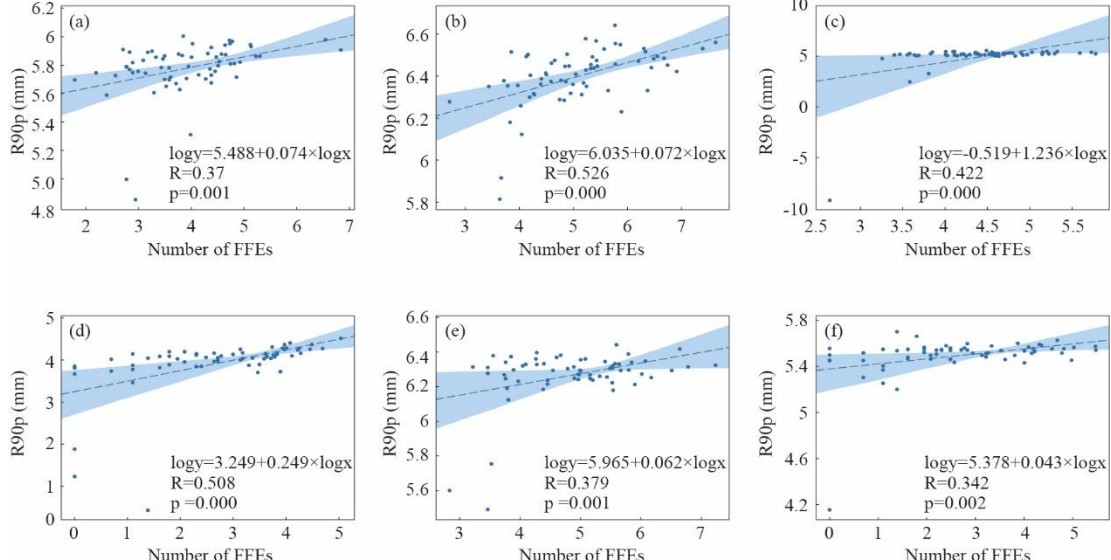

**Figure 11: Relations between R90p and FFEs. Where, R90p indicates the annual total precipitation amount of rainy days with precipitation exceeding 90th percentile. (a) EP; (b) SEM; (c); NCP; (d) NWB; (e) SWM; (f) TP.**




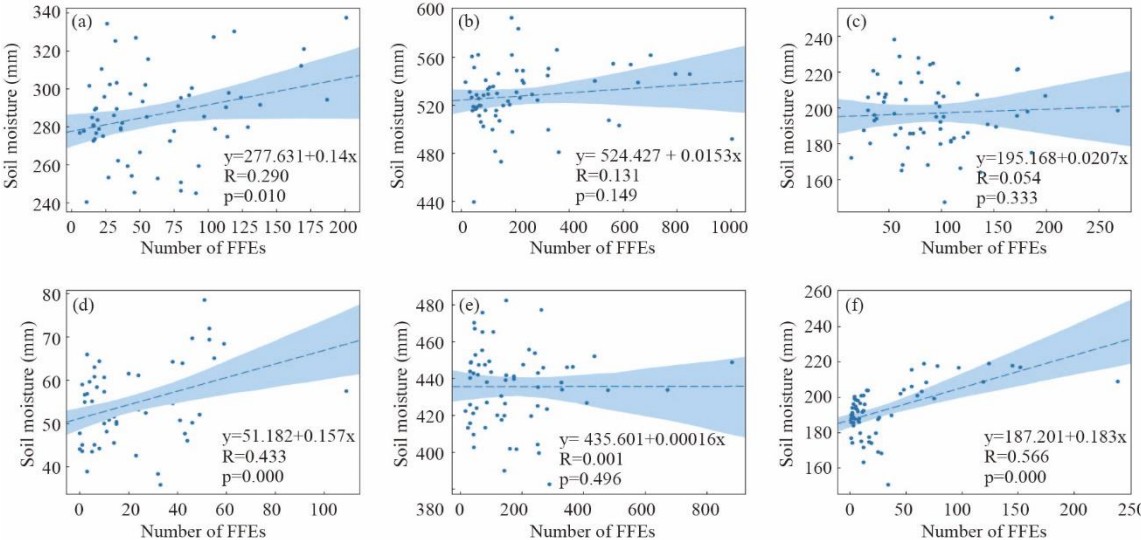

**Figure 12: Relations between the mean soil moisture of Summer (May–August) and FFEs. (a) EP; (b) SEM; (c); NCP; (d) NWB; (e) SWM; (f) TP.**

## 5.2 Typical peaks of the FFEs trending

In this study, a general increasing trend in frequency of FFEs was detected in the period 1950–2015, with more notable evidence in 2005–2010. Here, two obvious peaks, 1998 and 2010, were closely related to the extreme precipitation in China. To reveal the detailed relation between the extreme precipitation and FFEs in 1998 and 2010, the spatial patterns of these two elements were analyzed. According to the definition given by WMO, the average value of a meteorological element over 30 years is defined as a climatological normal. Therefore, the precipitation anomalies in this study was derived based on the mean precipitation of 1981–2010. Statistics results of the FFEs distributed in the different zones of the precipitation anomalies were listed in Table 4.

**Table 4** Distribution of the number of FFEs within the zone of the precipitation anomalies

| Anomalies of | 1998 | | | | | | 2010 | | | | | |
|---|---|---|---|---|---|---|---|---|---|---|---|---|
| precipitation (%) | May | Jun | Jul | Aug | Sep | Oct | May | Jun | Jul | Aug | Sep | Oct |
| <-40 | 1 | 14 | 25 | 48 | **15** | 2 | 1 | 45 | 22 | 46 | 2 | 1 |
| -40 ~ -20 | 6 | 9 | 75 | 42 | **13** | **3** | 15 | 21 | 55 | 32 | 2 | 0 |
| -20 ~ 0 | **30** | 39 | **209** | 29 | **9** | **3** | 22 | 32 | 66 | **54** | 19 | 24 |
| 0 ~ 20 | **30** | 53 | 143 | **68** | 7 | 0 | 19 | 60 | 130 | **144** | **31** | **170** |
| 20 ~ 40 | 13 | 60 | 137 | **97** | 4 | 0 | **31** | 65 | 294 | **52** | 24 | 119 |
| 40 ~ 60 | 6 | 75 | **149** | 57 | 1 | 0 | **28** | **103** | **381** | 28 | **43** | **184** |
| 60 ~ 80 | 9 | **107** | 133 | 67 | 1 | 1 | **31** | **161** | **457** | 11 | 13 | **283** |
| 80 ~ 100 | 5 | **231** | 95 | 31 | 2 | 0 | 6 | 92 | **417** | 5 | 11 | 77 |
| ≥100 | **21** | **239** | **261** | **140** | 0 | **3** | 5 | **110** | 613 | 6 | **25** | 32 |



In 1998, about 80% FFEs located in the zone with precipitation anomalies above 0 (Table 4). In August, the main rainfall belt was located over the Yangtze River, with precipitation 100% and more above normal. And a typical Secondary Meiyu occurred in August. Tropical and subtropical circulation systems were characterized by a stronger than normal and more westward-extending western Pacific subtropical high (WPSH), a weaker than normal EASM, and anomalous convergence of moisture flux in the mid and lower reaches of the Yangtze River. These similar circulation anomalies were attributed to the similar tropical sea surface temperature anomalies pattern in the preceding seasons, i.e., the super El Niño and strong warming in the tropical Indian Ocean (Yuan et al., 2017). The anomalies of precipitation resulted in the rainfall belt moving from lower to upper reaches along the Yangtze river from June to August. However, the FFEs occurred in September and October were not detected in the heavy rainfall center, which indicated there may be other factors together with the precipitation anomalies to induce the flash flooding during this period.

In 2010, more than 90% of the FFEs were distributed in the zone of the precipitation anomalies, with the zone of 60–80% and more than 100% ranking the top two (Table 4). Studies indicated that an El Niño Modoki with strongest warming in the central Pacific was detected, which caused the rainfall belt shifting northward. The extreme precipitation appeared in the Huaihe-Yellow River along with the weaken Indian summer monsoon and strengthen EASM (Wang et al., 2012). The cluster center of FFEs moved from Yangtze River to Huaihe River to Yellow River from May to July, which showed closely connection with the precipitation anomalies. Studies have indicated that the El Niño Modoki was different from the typical El Niño with respect to its evolution and climate impact (Feng et al., 2011).





**Figure 13: Percentage anomalies of precipitation averaged in May–August. Where, the red points are the location of FFEs. (a–f) May–August of 1998; (g–l) May–August of 2010.**



### 5.3 Deficiencies

In this study, we pointed out a few factors which could underlie the connection between the trending of FFEs and climate change. However, there is still a lack of definitive research into the driving factors of the temporal characteristics of FFEs. Additionally, the association detected between climatic factors and FFEs is not necessarily the causation of flash flooding
disasters. Our research has shown that currently there is not strong enough evidence to support whether climate change will do good or harm to the flash flooding disaster in the future. Besides, anthropogenic activities have imposed greatly complex impacts on the flash flooding disaster. On one hand, extensive reclamation of lakes and fluvial islands in the middle basin can considerably reduce the water storage and drainage capacity of watershed (Yin and Li, 2001; Zong and Chen, 2000). Meanwhile, the runoff generation potential may be improved due to the deforestation in the last decades (Khaleghi, 2017).
On the other hand, the increasing energy demands calls for more hydropower projects, which can reduce flood water during flood seasons (Bai et al., 2016). And the soil conservation implementation under the guidance of the Chinese government in recent years may reduce the peak flow in summer (Liu et al., 2019b).

### 6 Conclusions

To reveal the spatiotemporal characteristics of flash flooding events (FFEs) in China, the longest time series of FFEs was
analyzed to obtain the temporal variation, temporal periodic and temporal clustering on the national scale. Due to the huge diversity of China, a study to distinguish these detected trends were conducted in six geomorphologic regionalization (geomor-regions) during the period of 1950–2015, respectively.

1. The frequency of FFEs in China was detected to increase during 1950–2015, with more notable evidence in 2005–2010.The six geomor-regions showed the similar overall increasing trend, while each geomor-region displayed divergent patterns.
Two obvious peaks, 1998 and 2010, were identified to be closely related to the precipitation anomalies causing by the EASM and El Niño Modoki.

2. Approximately one third watersheds of all showed the significant upward trends in the period of 1950–2015. The most striking upward trends were detected in SWM, while the downward trend was only found in the east of EP. What's more, the changing magnitude of FFEs frequency were more significant in SEM and SWM.

3. On the large scale, 12–25a was identified as the main periodicity, with three oscillation periods, and tend to be stable since 1980. The most obvious temporal scale is 16–22a. On the small scales, the 2–8a time scale is prominent, with two oscillation periods, and tend to be stable since 2006, and 2–6a is another obvious temporal scale, which was concentrated after 2010. On the characteristic temporal scales of 19a, the average period was about six years, and the rich-poor periods alternate frequently with five prominent periodic oscillatory features.

4. The intra-annual trends in six geomor-regions showed the similar variation pattern, and the frequency distribution can be divided into right-skew, left-skew and symmetry. The inter-annual clustering played the dominant role in FFEs occurrence across China, while the under-dispersions were only detected in 5% out of all watersheds.

By and large, the research can be considered as a contribution towards advancing the acknowledge of spatiotemporal differentiation of the flash flooding disasters. For the diversity of China, the results obtained in this paper will be helpful in
gaining the better understanding of various generation mechanisms of flash flooding on the national scale. The findings of temporal pattern of FFEs in this study give the new insights on the temporal period, seasonality and clustering of FFEs response



to the global change in the long run. Besides, the results suggest that the temporal variations were closely related to the climatic variations and anthropogenic activities in China.

**Author contribution:** Nan Wang and Weiming Cheng conceived and designed the experiments; Nan Wang and Min Zhao
performed the experiments; Nan Wang and Qiangyi Liu analyzed the data; Nan Wang and Weiming Cheng wrote the paper; Jing Wang and Dongcheng Liu revised the paper.

**Acknowledgements:** This work was supported by the China National Flash Flood Disaster Prevention and Control Project. The authors are grateful for financial support from the China Institute of Water Resources and Hydropower Research (IWHR), grant number No. SHZH-IWHR-57 and National Natural Science Foundation of China, grant number No. 41571388.

**Conflicts of Interest:** The authors declare no conflict of interest.

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
