# Peer review of "Spatiotemporal analysis of flash flooding events in mountainous area of China during 1950–2015"

_Natural Hazards and Earth System Sciences, 2019_

## Referee Comment (RC1) · Anonymous Referee #1 · 4 Aug 2019

In this paper, the authors analyse the spatiotemporal characteristics of the flash flooding events (FFEs) observed in China for the period of 1950–2015, by using Mann-Kendall (MK) test, wavelet analysis, monthly frequency and index of dispersion.

General comments.

The authors try to evaluate the spatiotemporal characteristics of the FFEs in China by using a very large database, formed only by date and location of the events. No further information is available for each FFE, such has peak discharge or intense precipitation. The work is surely interesting as the timing features of FFEs have been extensively mapped, and provide a valid frame for further analyses on flash floods at smaller spatial and temporal scale. Anyway, often the obtained results cannot provide substantial information just for the intrinsic nature of the data and the too large areas

of the geomorphological regions, thus weakening some possible impacts of the work (such as enabling disaster estimation and prevention on the national scale, row 15). Classical statistical and mathematical methods have used to perform the analyses, nevertheless the results of some analyses are uncertain. Thus, many relevant issues need further deepening, as stated in the following.

First, in the title, the word "Statistical" may be added before "Spatiotemporal" to clearly indicate the approach followed in this study. Moreover, the authors specify that the analysis is focused to the mountainous areas of China, though in the text there is no mention at all about this fact. On the other side, the FFEs database is global, as shown in the figure 1. So, the call to the "mountainous areas" may be deleted, if not furtherly motivated in the text. The authors should provide further information about the specific criteria used for identifying the FFEs used in the analyses (e.g., distinguishing flash floods from normal floods). Moreover, a single climatic event may have caused different floods in a large watershed, or in conterminous smaller watersheds. Thus, within the database, the authors may have detected FFEs as floods caused by distinct climatic events (with different occurrence date), or as floods observed at different locations of a watershed at the same occurrence time. The reader may be confused in trying to understand the various analyses if this basic information is not clearly assessed. For the same aim, also the data aggregation used for the different analyses should be better defined all over the text (as correctly done in row 101).

As previously stated, the spatial scale of the study is very large, though the database is subdivided into six (not enough) smaller regions. This basic choice evidently weakens the search for relations between FFEs and climatic/physical features, such as rainfall and soil moisture that are generally locally varying variables. In fact, some results, only graphically visualized, show uncertain behaviours, probably due to peculiar features of the watersheds within the large regions. For example, this is the case of the skewness of the monthly frequencies (represented in figure 8 and in table 3), and of the regression analysis (represented in figures 11 and 12). To overcome partially the problem, the

visualization of the results through further, properly detailed, tables can be useful for comparing the statistical behaviours of the six different regions of China.

Some analyses show different behaviours within each geomor-region, that hardly can be averaged into a single specific behaviour. As an example, the intra-annual frequency distribution of FFEs has been divided into right-skew, left-skew and symmetry, but quite all the geomor-regions show great variability. As concern the temporal periodic analysis, the search for the inter-annual variation of the FFEs should be performed at the geomor-region space scale, and then physically explained.

From a formal point of view, the work is well structured, and it is based on a huge database, valuable for a statistical analysis. The subject is very interesting, and the text should be rewritten only in some specific section for the sake of clarity. The presentation of the results has to be improved in some parts. The list of references is exhaustive and well chosen. The quality of the figures is high, but some of them should be substantially improved. All the suggestions/corrections proposed for improving the text are listed in the following specific comments.

As a result of the review, I recommend major corrections for this manuscript before publication on Natural Hazard and Earth System Sciences.

Specific comments.

Some requests on specific topics are listed below. - Section 2.3. The subdivision of the entire study area into 133 watersheds based on third-order stream is not a completely exhaustive information on their features. In fact, the large range of the watersheds area needs some further explanations that can be inserted in table 1. - Section 3.1. The two-tailed test for the Sen's slope can be better defined. - Section 4.1. At the start of the section 4.1, before presenting the results, it is useful a short mention to the method used for assessing the significance of the trends. Moreover, the mean annual change rates of the various regions (rows 171-173) do not coincide with the slope of the regression equations showed in figure 2. In order to increase readability,

the authors should represent the regression equations as FFEs=a+b*(current year-1950). Moreover, the results evidenced high intensities in two phases (1985–1998 and 2000–2010) for most regions. Can the authors discuss about the probable reasons of the lower number of FFEs before 1985 (lower quality/quantity of information, lower precipitation . . .)? - Sections 4.1 and 4.2 need some further explanation as regards the database (data and location of FFEs) used for analyses. Section 4.1 copes with the trend in the FFEs global number series of each geomor-regions, while section 4.2 deals with the different watersheds of the geomor-regions. This different data aggregation should be better defined; otherwise, it can be easily misunderstood. Moreover, control the pertinence of the word "mutation" in the title of section 4.2. - Section 4.3. This section try to find oscillation periods of the FFEs database at large and small scales, but the global result shows a complex timing frame that should be adequately explained. The authors are requested to relate the potential periodic features to some external (physical, climatic or planetary) factors, which could reinforce this potential result. On the other side, the wavelet analysis has been referred to the whole China, while in the previous analysis the six geomor-regions have shown peculiar behaviours. Why did not the authors perform this analysis to the FFEs database of the different geomor-regions? Can the authors try to match this result with the two peaks, 1998 and 2010, closely related to the precipitation anomalies caused by the EASM and El Niño Modoki? - 4.4.1 and 4.4.2 have the same titles. - Subsection 4.4.1. The results showed in rows 253-258 cannot be easily related to figure 8, which contains different coloured lines. Probably, the authors presents the results related to the mean monthly frequency of each geomor-region. If my understanding is true, some uncertainties appear in the results (and in table 3). For example, the NWB region also shows a symmetry distribution like the SEM and SWM regions. If not, authors should help the reader in understanding the presented results. Moreover, the worthy attempt to relate behaviour of FFEs with seasonality of precipitation can be further improved. - Section 5. The title of the subsection 5.1 is not good. The section contains regression analyses between number of FFEs and some potential physical factors (not impacts) which may induce

flash floods. - Subsection 5.1 can be improved by writing it in a more clear way, in order to avoid repetitions and uncertainties (see notes for row 305 and figure 11). Moreover, the relations between number of FFEs and physical factors (precipitation index, soil moisture) are very uncertain, due to the large spatial scale of the study. The authors, when necessary all over the text, should better stress this fundamental point.

Technical corrections.

Text. - Row 18. "Periodic" is an adjective not a noun. - Row 117. Change "Where" with "where". Add the information "i<j". - Row 121. Change "Formula" with "the following formula". - Row 123. Change "Where" with "where". Change "vise" with "vice". - Row 135. Change "Where" with "where". - Rows 135-136. The sentence "here, Morlet wavelet was chosen as the mother wavelet function" may be inserted within round brackets. - Row 146. Change "Where" with "where". - Rows141-142. Move the sentence "However, if a monthly . . . 5% significance level" at the end of the row 149. - Row 157. Change "Where" with "where". - Row 168. Separate the words "Figure 2c". - Row 169. Separate the words "Figure 2f". - Rows 174-176. The increase of the number of FFEs for NWB is vaguely defined (speeding rate), though the NWB and TP regions seem to have the same exponential behaviour. - Row 179. Separate the words "yellow line". - Row 184. Separate the words "Figure 3". Change "33% watershed of all" with "33% of all watershed". - Rows 186-188. The sentence is not clear, rewrite in a better English style. - Row 187. Separate the words "Figure 3". - Row 189. Change "southeast" with "southeastern". - Row 200 (caption of figure 3). Change "Where" with "where". - Row 208. Separate the words "Figure 4". - Row 209. At this first appearance, define the meaning of the symbol "Xa", where X is a number. - Row 224. Change "were" with "are". - Row 225. Change "corresponded" with "correspond". - Rows 227-229. The sentence is not clear, rewrite in a better English style. - Row 234. Separate the words "Figure 7". - Row 247. Separate the words "Figure 8". - Row 250 (caption of figure 8). Delete "Where". - Rows 254-255. The sentence is not clear, rewrite in a better English style. - Row 274. Separate the

words "Figure 10". - Row 278. Do the authors intend "regular distribution of monthly FFEs occurrence"? - Rows 290-291. The definition of the factor R90p is not clear (cumulative precipitation formed only by daily precipitation greater than 90th percentile of 1980-2010 precipitation?). Moreover, here it is indicated as seasonal precipitation, while caption of figure 11 indicates R90p as annual total precipitation. - Row 293. Separate the words "Figure 11". - Row 304. How is assessed the statistical significance of the correlation? - Row 305. The importance of the role of the soil moisture on FFEs has been assessed before in row 289. - Row 330. The word "similar" is repeated in the same sentence. - Row 334. It is not clear what is the heavy rainfall center. Maybe it has to be related to figure 13 (not cited)? Moreover, change "center" with "centre". - Row 344 (caption of figure 13). Delete "Where". - Row 356. Do not start a sentence with "And". - Row 360. "Periodic" is an adjective not a noun. - Row 367. Delete the words "of all" or change the sentence. - Row 381. Change "the new insights" with "new insights".

Figures. - Figures 1 and 3. Change the NWP symbol into NWB (as in the description of the six geomorphologic regions of table 1). - Figure 2. The equations represent the relationships between the year and the number of FFEs for each regions. The suggestion is to change regression equations as previously suggested for rows 171-173. - Figure 4. At the end of the caption, add the words "for the entire China". - Figure 5. After the words "of FFEs" in the caption, add the words "for the entire China". Delete the article "the" at the start of the parts (a) and (b) of the caption. - Figure 6. At the end of the caption, add the words "for China". - Figure 8 seems not to agree with figure 1c, even if this can be a trivial problem of scale representation. In fact, some geomor-regions of figure 8 show regional FFEs values in months like February and November, which have no concordance in figure 1c. Can the authors provide an explanation? - Figure 9. The caption can be shortened deleting the words "Maps showing the" and "Where". - Figure 11. The graphs are in logarithmic scale; therefore, both the labels of the variables have to be indicated with logarithm. Moreover, the suggestion to be clearer in the definition of variables all over the work here is fundamental. In fact,

it is not clear what the points could be (Number of FFEs for each location within a watershed of a region?). In other words, while the red points of figure 1 are obviously all the FFEs collected for this work, it is not clear what the points of figures 11 and 12 really could be. - Figure 13 is not cited in the text. The six graphs for each year do not correspond to the four months May-August of the caption.

Tables. - Table 2. For the sake of readability, add the total number of watersheds for each regions. This can be very useful for a better understanding of the following analyses.

---

## Author Comment (AC1) · 26 Aug 2019

The comment was uploaded in the form of a supplement:
https://www.nat-hazards-earth-syst-sci-discuss.net/nhess-2019-150/nhess-2019-150-AC1-supplement.zip

---

## Short Comment (SC1) · 5 Sep 2019

1. This paper has a strong contribution in the study of spatiotemporal variation and trend of flash floods at a national scale, and therefore fits the goal and objective of the journal. It also has an importance in risk assessment of flash floods in China under climate change conditions. 2. An emphasis on the importance of this research can be added in the abstract. 3. A short review of similar studies at different scales in China or other nations can be added in Section 1 if any. 4. The objectives of this study need to be clearly addressed in Section 1. 5. Section 2.1: Not clear how these FFEs were selected, by peak or by damage? Needs to have more descriptions on the criteria of flash floods at different scales. 6. Section 2.3: Watershed was used for calculating the temporal mutation and clustering. Were there a condition that the FFE locations in the

dataset have an upstream-downstream connection within the watershed? If this was the case, would it affect the assumption that FFEs were independent for the statistical analysis. Needs to have a clarification or discussion on this. 7. Better to give an equation or reference for wavelet variance in Figure 6 and wavelet coefficient in Figure 7 for an easy understanding. 8. A more description or equation is suggested for the index of dispersion in Section 4.4.2 and Figure 10. 9. Section 5.1: What does the light blue area in Figure 11 and Figure 12 stand for? How was the R90p calculated for a geomor-region? Is it possible to have a negative R90p in Figure 11(c) (a point in the left bottom corner)? How was the soil moisture estimated for a region? To what soil depth? Suggest to have a discussion on different soil moisture magnitudes in different regions, e.g. Figure 12(d) and Figure 12(e). 10. Figure 13: have a check of the figure title, four maps for each year (May–August of 1998; May–August of 2010), but 12 maps are in the figure. 11. Section 5.3: A brief discussion can be conducted on the influencing factors of flash floods which were not included in the analysis, for instance, vegetation cover, drainage area, landscape, soil texture, and geological conditions. 12. English can be further improved.

---

## Author Comment (AC2) · 19 Sep 2019

The comment was uploaded in the form of a supplement:
https://www.nat-hazards-earth-syst-sci-discuss.net/nhess-2019-150/nhess-2019-150-AC2-supplement.zip

---

## Referee Comment (RC2) · Anonymous Referee #2 · 1 Nov 2019

This study uses a very interesting data set about flood event across China for the last 75 years. It analyses spatio-temporal patterns and attempts to link them to rainfall and soil moisture. Although I think that this data set may offer a great opportunity and that the research question is of high importance, the manuscript contains flaws, does not given the information to understand the methods and results and is not written in a concise way.

Major Comments:

(1) Manuscript lacks conciseness: The manuscript is hard to understand as it lacks conciseness. For example, sentences like "...Precipitation anomalies and soil moisture were detected to have a close correlation with FFEs, however, the interplay of climatic variations and anthropogenic activities may impose greatly impacts on the occurrence

and evolution of the flash flooding disasters on a large extent. . ." leave the reader with an uneasy feeling as it is not completely clear what is meant by this sentence and by some of its parts (such as ". . . on a large extent. . ."). On many locations, the reader knows what is meant, but still it is not concise. An example (Line 42): ". . .driven by increasing precipitation and atmospheric circulation. . .": I assume that it is not increasing atmospheric circulation but changes in the frequency and persistence of flood-related circulation patterns. Another example is Line 77: ". . .including temporal variation, temporal mutation, temporal periodic, and temporal clustering. . .": Is temporal variation different from the 3 other terms? What is meant by temporal mutation and temporal periodic? The manuscript also contains statements that are not substantiated and may mislead the reader. For example: ". . .Our research has shown that currently there is not strong enough evidence to support whether climate change will do good or harm to the flash flooding disaster in the future. . ." I would rather argue that this manuscript has not addressed this question. There is a large body of literature and very elaborated methods on attributing changes (for example in rainfall or flooding) to climate change. Hence, saying that an important question cannot be answered as there is not enough evidence is inappropriate, when the research has not even attempted to address the attribution question. Another example for unconcise wording: ". . . Besides, the results suggest that the temporal variations were closely related to the climatic variations and anthropogenic activities in China. . .": Anthropogenic activities are mentioned in a very general way, and the manuscript does not contain any analyses or specific statements about anthropogenic activities. Unfortunately, these are only a few examples.

(2) Manuscript contains a number of errors: An example is the caption of Figure 1: (b) is the trend and not the intra-year series, and (c) is the intra-year distribution and not the inter-year series. Another example: Sections 4.4.1 and 4.4.2 have exactly the same title (Intra-annual clustering), but have different contents.

(3) The English is poor, and often very difficult to understand.

(4) Structure of the manuscript: The discussion chapter takes up a new issue and

introduces additional data (precipitation and soil moisture). I strongly recommend to rearrange the contents such that all data, methods and results are reported in the data, methods and results sections, respectively, and that the discussion section only discusses the findings.

(5) Abstract: The abstract cannot be fully understood since terms are used which can have different meanings, but are not explained.

(6) I am confused by the use of the term "Flash Flooding". Flash floods are different from river floods. In the introduction, several "flash flood" studies are cited, but these studies actually discuss river floods and not flash floods. First, I thought that the authors had not carefully read the literature they were citing, but later I got very confused as they speak about large watershed ("... 133 watersheds, with the watershed area ranged from 0.3 to 60 ×104 km2 ..."). Is this paper about flash floods or river floods? I also propose that they introduce their definition of flash flood events early in the paper.

(7) Selection of Events and FFE database (Section 2.1): The criteria when an event has been counted/documented in the database are not given. The data set needs to be described in detail. For example, has been taken care of the reporting bias? Without a detailed describtion, the reader cannot really interpret the results of the study.

(8) Division into regions (Section 2.2): The justification for dividing the country into these 6 regions is not given. Why these 6 regions? Are 6 regions detailed enough for such a large country?

(9) Description of data is incomplete (Section 2.4): A much more detailed description of the data used is necessary. For instance, how many rainfall stations are available? What is the (grid) resolution of the simulated soil moisture?

(10) Section 2.4: Rainfall data is only available for 1980–2010, but the event FFE analysis is carried out for 1950-2015. What is done for the periods where rainfall data is missing?

(11) Figure 1: It is not clear what Figure 1c shows. What is the meaning of the blue polygone?

(12) Methods: Different methods are used but only for the trend analysis with Sen's slope, the significance is calculated. I feel that it is absolutely necessary to provide significance statements also for the other analyses (reported in 4.1, 4.3, 4.4).

(13) Methods/results: Figure 2 shows ". . . trend line based on the least-squares linear regression. . .". This is inconsistent with the Methods Section where it is stated that the trend is estimated by Sen's slope.

(14) Methods/results: The grey, yellow and blue lines seem to decrease to 0 at the end of the time series. I guess that this is not a real decline in the occurrence of floods (then the red line would not increase), but is an artefact of the method/presentation used. This needs to be clarified.

(15) Method (Line 192): The sentence "The statistic results presented an overall agreement between MK testing and Sen's slope estimator. . ." is unclear (as MK tests the significance and Sen provides the slope of the trend).

(16) Section 4.1 and 4.2: Both sections present trend analyses. Why are there 2 sections with different methods looking at the trend? I rather feel that the reader is confused by these 2 sections (in particular as the section titles are not understandable).

(17) Figure 3: The value of the trend (Sen's slope) should be given in relation to a certain quantity, e.g. X% per decade in relation to average value. Otherwise, these numbers do not give any information. Similarly, I wonder whether the reader can interpret the Z-statistics of the MK test. I would rather report the p-value which is typically used.

(18) Figure 4/Figure 5: The wavelet results look quite different to the many wavelet results I have seen in other studies. I propose to use the setup of the seminal paper on wavelets of Torrence and Compo (1998). Further, one needs to add the cone of

influence to understand where the results are not reliable. In addition, I wonder why 3 wavelet diagrams are shown (Fig. 4, 5a, 5b). Why do they differ?

(19) Wavelet results: I wonder why the annual scale is not prominent. Figure 1c seems to suggest that there is a very strong seasonality in the occurrence of FFEs; then this should show up in the wavelet results. (But maybe I misinterpret Figure 1c, because it is not explained what this figure (blue polygone) means).

(20) Figure 7: I understand that Figure 7 shows a transects at the main periods (10a, 5a) through the wavelet diagram (Figure 4), but I cannot understand at all the explanation of this figure (Lines 233-238).

(21) Methods: Intra-annual clustering (Section 4.4.2): It is not clear what is shown and discussed in Figure 10 and in section 4.4.2. For example, the calculation of D requires selecting the aggregation period T; this information is not given. What exactly is meant by intra-annual clustering? Further, significance of clustering derived by the index of dispersion method is sensitive to the selection of the starting time point of the aggregation window (see Merz et al., 2016). This issue should be considered.

(22) Figure 11 and 12: I assume that 'number of FFEs' means the total number of documented FFEs within each year (from Jan to Dec). This needs to be said. More disturbing is that the x-axis of Figure 11 seems to have wrong labels as the numbers are 2 orders of magnitude smaller than Figure 12.

(23) Figure 13: The text does not contain a hint to and discussion of Figure 13. However, this is necessary, as it is not that obvious from Figure 13 that FFEs occurrence and high positive anomalies match.

Minor Comments:

(24) Line 19: Please relate the growth rate ("... with a growth rate of 23.62 per year..." to the average number per year or give the growth rate in %. Otherwise its relevance cannot be understood.

(25) Line 20, 21: Please specify what is meant by ". . . large scale. . .", ". . . small scale. . .".

(26) Line 22: The sentence: ". . . intra-annual frequency distribution of FFEs can be divided into three types, right-skew, left-skew and symmetry;. . ." cannot be understood; at this point it is not clear what is meant by intra-annual frequency distribution.

(27) Line 23: Again, it is unclear what is meant by ". . . inter-annual clustering . . ." and ". . . under-dispersions. . .".

(28) Line 35: ". . .impacts of global change on climate, severe weather in the form of heavy rainfall and river discharge conditions . . .": What exactly is meant here? Are the second and third causes not consequences of the first cause? Or do you mean that surface processes change (independent of the climate)?

(29) Line 39: This sentence is unclear: ". . .Most previous studies related to the multi-variate frequency analysis of extreme events assumed temporal stationarity. . .." Firstly, I do not understand what is meant by multivariate frequency analysis; secondly, I think that there are many papers meanwhile that have not assumed stationarity.

(30) Line 49: ". . .However, few studies have been focused on the spatiotemporal changing of flash flooding on the national scale in China. . .": Please list these studies here.

(31) Line 58: What is meant by ". . .of the flash flooding intra annual clustering. . ."?

(32) Line 59: What is meant by ". . .most probable flash flooding generation processes can therefore assist in the identification of homogeneous regions with a dominant flash flooding season. . ."?

(33) Line 67: What are ". . .geomor-regions. . ."?

(34) Line 127: What exactly is meant by ". . . Sen's slope is tested by a two-tailed test at $\alpha$ confidence. . ."? Is this sentence related to the Mann-Kendall test mentioned earlier?

(35) Table 4: Please explain in the caption the meaning of the colours in the table.

[Figure]

(36) Line 337: Please explain the term 'El Niño Modoki'.

---

## Author Comment (AC3) · 12 Dec 2019

Dear Editors and Reviewer,

Thank you very much for your messages and your efforts in processing our manuscript "Spatiotemporal analysis of flash flooding events in mountainous area of China during 1950–2015" (MS No.: nhess-2019-150). My colleagues and I are very grateful to you for the valuable comments. Based on them we have revised the paper which is attached for your further consideration. Please refer to the enclosed "Responses to the referee #2's comments" for details on the substantial revisions we have made. Our responses are right after each comment.

Please feel free to contact us if you have any questions with our revision of this paper.

Sincerely, Nan Wang

Institute of Geographic Sciences and Natural Resources Research, Chinese Academy of Sciences, 11A, Datun Road, Chaoyang District, Beijing, China Tel.: +86 13011870541 Email: wnan@lreis.ac.cn

**Responses to the referee #2's comments**

**• General comments**

**Comments:** This study uses a very interesting data set about flood event across China for the last 75 years. It analyses spatio-temporal patterns and attempts to link them to rainfall and soil moisture. Although I think that this data set may offer a great opportunity and that the research question is of high importance, the manuscript contains flaws, does not given the information to understand the methods and results and is not written in a concise way.

**Response:** Thank you very much for your time on our manuscript and the opportunity to revise the work. We took these comments and suggestions seriously and addressed them in every detail we could. We hope the revisions can help you and the readers to understand the methods and results more easily. We have carefully modified the language in the original manuscript to make it in a more understandable and concise way. The key points of the revision are as follows:

- 1) The writing has been improved and checked throughout the whole manuscript. The errors in the pictures and table has been corrected.
- 2) The structure of the manuscript has been modified and the Dataset, Results and Discussion sections have been separated.
- 3) The confused items and statements have been clarified to better explain our methods and finding in the spatiotemporal patterns of flash flood disasters.
- 4) Some supplement to the method has been added to make the results more credible.

**Major comments**

**Comments:** Manuscript lacks conciseness: The manuscript is hard to understand as it lacks conciseness. For example, sentences like ". . . Precipitation anomalies and soil moisture were detected to have a close correlation with FFEs, however, the interplay of climatic variations and anthropogenic activities may impose greatly impacts on the occurrence and evolution of the flash flooding disasters on a large extent. . ." leave the reader with an uneasy feeling as it is not completely clear what is meant by this sentence and by some of its parts (such as ". . . on a large extent. . .").

**Response:** Thank you for your kind comments and suggestions. We checked throughout the whole manuscript to make it more concise.

**Comments:** On many locations, the reader knows what is meant, but still it is not concise. An example (Line 42): ". . .driven by increasing precipitation and atmospheric circulation. . .": I assume that it is not increasing atmospheric circulation but changes in the frequency and persistence of flood-related circulation patterns.

**Response:** Thank you for your kind comments and suggestions. We modified the description in the revised version.

**Revised** (Lines 74-76 in the revised version): A clear upward trend in flood frequency in Germany was proven to be driven by extreme precipitation and atmospheric circulation (Petrow and Merz, 2009).

**Comments:** Another example is Line 77: ". . . including temporal variation, temporal mutation, temporal periodic, and temporal clustering. . .": Is temporal variation different from the 3 other terms? What is meant by temporal mutation and temporal periodic?

**Response:** Thank you for your kind comments and suggestions. Firstly, we agree with you that the term "temporal variation" is somewhat confusion, therefore, we changed it into "temporal evolution". Temporal evolution was intended to analyse the long-term time series based on the annual number of FFEs in each geomor-region. In addition, "temporal mutation" was changed into "temporal trends".

Temporal trends were supposed to detect the upward/downward trends of the occurrence of FFEs in each watershed. It can also reveal the distribution of the divergent temporal trends in spatial. Finally, "temporal periodic" was changed into "temporal period", which was used to detect whether there were predominant periods at different scales or not in the temporal evolution of FFEs.

**Revised (Lines 108-110 in the revised version):** This is followed by the analysis of spatiotemporal characteristics of FFEs, including temporal evolution, trends, period, and clustering in Section 4.

**Comments:** The manuscript also contains statements that are not substantiated and may mislead the reader. For example: ". . .Our research has shown that currently there is not strong enough evidence to support whether climate change will do good or harm to the flash flooding disaster in the future. . ." I would rather argue that this manuscript has not addressed this question. There is a large body of literature and very elaborated methods on attributing changes (for example in rainfall or flooding) to climate change. Hence, saying that an important question cannot be answered as there is not enough evidence is inappropriate, when the research has not even attempted to address the attribution question.

**Response:** Thank you for pointing out the not rigorous statements in the original manuscript and we have corrected them. In this paper, we intended to induce the statement that the climate change might has a complex impact on the flash floods. But we did not have strong enough evidence to support these statements.

**Revised** (Lines 578-579 in the revised version): Our research has not provided strong enough evidence to support whether climate change will mitigate or propagate flash flood disasters in the future.

**Comments:** Another example for unconcise wording: "... Besides, the results suggest that the temporal variations were closely related to the climatic variations and anthropogenic activities in China. . .": Anthropogenic activities are mentioned in a very general way, and the manuscript does not contain any analyses or specific statements about anthropogenic activities. Unfortunately, these are only a few examples.

**Response:** Thank you for the insightful comments. We have deleted the unconcise wording in the revised version.

**Comments:** Manuscript contains a number of errors: An example is the caption of Figure 1: (b) is the trend and not the intra-year series, and (c) is the intra-year distribution and not the inter-year series. **Response:** Thank you for the detailed comments. We have checked throughout the figures and tables in the manuscript and corrected the errors.

**Revised (Figure 1 in the revised version):**

**Figure 1:** Location and inter-year and intra-year series of FFEs in China over 1950-2015. (a) the spatial location of the study area and the distribution of FFEs; (b) the inter-year series of FFEs; (c) the intra-year series of FFEs.

**Comments:** Another example: Sections 4.4.1 and 4.4.2 have exactly the same title (Intra-annual clustering), but have different contents.

**Response:** Thank you for the comments. We have modified the titles of Sections 4.4.1 and 4.4.2. **Revised (Lines 339 and 388 in the revised version):** 4.4.1 Intra-annual clustering; 4.4.2 Inter-annual clustering

Comments: The English is poor, and often very difficult to understand.

**Response:** Thank you for the comments and we improved the English writing style to make it more understandable.

**Comments:** Structure of the manuscript: The discussion chapter takes up a new issue and introduces additional data (precipitation and soil moisture). I strongly recommend to rearrange the contents such that all data, methods and results are reported in the data, methods and results sections, respectively, and that the discussion section only discusses the findings.

**Response:** Thank you for the insightful suggestion. According to your suggestion, we compiled all data used in this study in the Section 2. In the discussion, we focused on the more notable period with the high occurrence of FFEs. Besides, we attempted to analyze some of the important factors which may explain the reason for the temporal variation of FFEs.

**Comments:** Abstract: The abstract cannot be fully understood since terms are used which can have different meanings, but are not explained.

**Response:** Thank you for the suggestion. We modified the abstract and used the more understanding terms to make it more clear.

**Revised (Lines 17-36 in the revised version):** To bridge this gap in the research on the spatiotemporal characteristics of flash flood events (FFEs), this study used a Mann-Kendall (MK) test, wavelet analysis, monthly frequency, and index of dispersion (D), based on the longest time series available of FFEs in China to detect the temporal evolution, trends, period, and clustering of FFEs in six geomorphological regions (geomor-regions: the EP, SEM, NCP, NWB, SWM, and TP) of China. The results indicated that: (1) the frequency of FFEs in the EP and NCP regions increased steadily with rates of change of 3.34 and

1.86 per year and at rates of 9.32 and 8.05 per year in the SEM and SWM, and increased dramatically in TP and NWB in the last twenty years; (2) the watersheds with upward trends at the 99% significance level were mainly located in the SEM and SWM regions, while those with downward trends at the 90% significance level were only detected in the EP region; (3) in EP, SEM, and SWM, there were three clear oscillation periods on the time scale of 12–25a and two clear oscillation periods on the scale of 2–8a; (4) the highest monthly frequency of FFEs was more likely to occur in July, and it appeared to occur after July in EP, NCP, and TP and before July in SEM and NWB; (5) The inter-annual clustering of FFEs played a dominant role across China, while the typical pattern of FFE occurrence was only detected in six (five percent) watersheds.

**Comments:** I am confused by the use of the term "Flash Flooding". Flash floods are different from river floods. In the introduction, several "flash flood" studies are cited, but these studies actually discuss river floods and not flash floods. First, I thought that the authors had not carefully read the literature they were citing, but later I got very confused as they speak about large watershed (". . . 133 watersheds, with the watershed area ranged from 0.3 to  $60 \times 10^4 \text{ km}^2 \dots$ "). Is this paper about flash floods or river floods? I also propose that they introduce their definition of flash flood events early in the paper.

**Response:** Thank you for the thoughtful comments. Firstly, for the term "flash flooding" we used in this paper maybe not so widespread, we changed the "flash flooding" into "flash floods" throughout the paper. Secondly, the flash floods analysed in this paper included the water floods, debris flows and landslides etc., which occurred in the mountainous area and caused the economic losses or the people death. The flash flood events we discussed in this paper is comprised of several kinds of disasters triggered by high-intensity and short-duration rainfall (or snowmelt). Therefore, the studies we cited in Section 1 included river floods and flash floods. The watersheds were used as the basic units to sum up the number of FFEs in the analysis of monthly frequency, index of dispersion, and the relationship between climatic factors and FFEs. According to your kindly suggestion, we mentioned the definition of flash flood events in Section 1, and we also added some description of the FFEs inventory in Section 2 to make it more understandable for readers.

**Revised (Lines 120-122 in the revised version):** The FFEs analysed in this paper included water floods, debris flows, and landslides which occurred in mountainous areas and caused economic losses or human fatalities.

**Comments:** Selection of Events and FFE database (Section 2.1): The criteria when an event has been counted/documented in the database are not given. The data set needs to be described in detail. For example, has been taken care of the reporting bias? Without a detailed description, the reader cannot really interpret the results of the study.

**Response:** Thank you for the useful suggestion. The FFEs in this study were obtained by the official documents and the published literature, which were considered as the trustful information sources. What's more, some interview and field work have been done to deal with the records which have no detailed description. In the FFEs database, each FFE was labelled with the unique PID, which indicated the index of the administrative village. In this study, we have also dealt with the repeated records with the same PID and the same detailed information to avoid the repetitive records.

**Revised (Lines 122-124 in the revised version)**: In addition, we have eliminated repeated instances of FFEs from the data to ensure that each event is recorded only once in the same administrative village (which is the smallest administrative unit in China) and at the same date.

**Comments:** Division into regions (Section 2.2): The justification for dividing the country into these 6 regions is not given. Why these 6 regions? Are 6 regions detailed enough for such a large country? **Response:** Thank you for the comments. The geomorphological regionalization (geomor-region) we used in this paper were widely acknowledged by researchers for the division of China into some subregions. We analysed the spatiotemporal characteristics of FFEs in six geomor-regions in China based on the following points. Firstly, the topographic features of watersheds, including slope, aspect, relief, etc., have an important impact on the occurrence of flash floods. Secondly, the watersheds within the same geomor-region tend to share the same or similar climatic conditions. Thirdly, the six geomorregions showed distinctive characteristics in terrain, climate, soil, vegetation, together with population distribution, which are widely recognized by researchers. For the huge divergent of China, the spatiotemporal analysis of FFEs was conducted in six geomor-regions respectively can reveal more detailed information. However, if we adopted a more detailed division, there will not be sufficient FFE samples for a complete long-term record in each detailed division. Therefore, after the comparison and consideration, we selected the six geomor-regions as the proper division of spatiotemporal analysis of FFEs.

**Revised (Lines 134-135 in the revised version):** Accordingly, based on the regional differentiation of essential geomorphologic types and their genesis, the entire country has been divided into six major geomor-regions (Table 2, Figure 1).

| Name                 | Abbreviation | Description                                                                |  |  |  |  |
|----------------------|--------------|----------------------------------------------------------------------------|--|--|--|--|
|                      |              | This region is in the northern part of China. It comprises low terrains    |  |  |  |  |
|                      |              | and the largest plain areas. Plains and platforms are dominant features    |  |  |  |  |
|                      |              | of this region, along with well-developed fluvial accumulation             |  |  |  |  |
|                      | EP           | landforms. Major watersheds in this region include the Songhua River       |  |  |  |  |
| Fastorn Hilly Dising |              | Watershed, the Tumen River Watershed, and the Huai River Watershed.        |  |  |  |  |
| Eastern milly Plains |              | Flash floods in the Songhua River watershed are mostly caused by           |  |  |  |  |
|                      |              | heavy storms, 80% of flash floods occur in July and August (especially     |  |  |  |  |
|                      |              | in August). Flash floods in the Huai River Watershed can be caused by      |  |  |  |  |
|                      |              | Meiyu and heavy storms, and the flash floods are generally concentrated    |  |  |  |  |
|                      |              | from June to September.                                                    |  |  |  |  |
|                      | SEM          | This region is located in the southern part of the low terrain topography, |  |  |  |  |
|                      |              | and it is dominated by low elevation hills and low or middle relief        |  |  |  |  |
| Southoastern Low     |              | mountains, with only 30% of its area occupied by plains and platforms.     |  |  |  |  |
| middle Mountains     |              | Major watersheds in this region include the middle and lower reaches of    |  |  |  |  |
| inidale Mountains    |              | the Yangtze River Watershed and the Poyang Lake Watershed. Flash           |  |  |  |  |
|                      |              | floods are the most frequent and serious in middle and lower reaches of    |  |  |  |  |
|                      |              | the Yangtze River Watershed because of the Meiyu and heavy storms.         |  |  |  |  |
|                      |              | This region is located in the north-eastern part of China's middle terrain |  |  |  |  |
|                      |              | topography and is characterised by a plateau landform composed of low      |  |  |  |  |
|                      |              | or middle relief mountains, hills, platforms, and plains. The loess        |  |  |  |  |
| Northern and Central |              | landform is well developed in this region. Major watersheds in this        |  |  |  |  |
| Middle Mountains     |              | region include the middle reaches of the Yellow River Watershed and        |  |  |  |  |
| and Plains           |              | the Wei River Watershed,. Ice floods are easily caused by ice jams, ice    |  |  |  |  |
|                      |              | dams, etc. in the middle and lower reaches of the Yellow River from        |  |  |  |  |
|                      |              | December to March. Flash floods also concentrate in late July to early     |  |  |  |  |
|                      |              | August and are caused by heavy storms.                                     |  |  |  |  |
|                      |              | This region is located in the north-western part of the middle terrain     |  |  |  |  |
|                      |              | topography. It is composed of middle to high mountains with flattened      |  |  |  |  |
| North-western        | NWB          | basins interposed between them, and it is characterised by an arid desert  |  |  |  |  |
| Middle and High      |              | geomorphology. Mountains with basins are made up of plains,                |  |  |  |  |
| Mountains and        |              | platforms, and hills. Major watersheds in this region include the Tarim    |  |  |  |  |
| Basins               |              | River Watershed and the Ili River Watershed. Flash floods in this region   |  |  |  |  |
|                      |              | can be caused by local heavy storms or the combination of heavy storms     |  |  |  |  |
|                      |              | and snowmelt.                                                              |  |  |  |  |
| Southwestern         | SWM          | This region is located in the southern part of the middle terrain          |  |  |  |  |

Table 2 Description of the six geomor-regions in China

| Subalpine Mountains |    | topography. Evidencing a typical karst landform, middle or high             |
|---------------------|----|-----------------------------------------------------------------------------|
|                     |    | mountains with middle or high reliefs are widespread with wide valley       |
|                     |    | basins interspersed between them. Major watersheds in this region           |
|                     |    | include the upper and middle reaches of the Yangtze River Watershed         |
|                     |    | and the Jialing River Watershed. Influenced by the plateau monsoon          |
|                     |    | climate, the storm period in this region is long and usually multi-peaked.  |
|                     |    | This region encompasses China's high terrain topography. It is              |
|                     |    | composed of plains and high mountains at elevations above 4,000 m           |
|                     |    | which account for three-fourths of the area of the region. It is            |
| Tibetan Plateau     | TD | characterised by glacial and periglacial landforms. Major watersheds in     |
|                     | 1P | this region include the Yarlung Zangbo River Watershed, Nu River            |
|                     |    | Watershed, and Shiquan River Watershed. Local persistent heavy rain         |
|                     |    | are the main cause of flash floods in the tributaries of the middle reaches |
|                     |    | of the Yarlung Zangbo River Watershed.                                      |

**Comments:** Description of data is incomplete (Section 2.4): A much more detailed description of the data used is necessary. For instance, how many rainfall stations are available? What is the (grid) resolution of the simulated soil moisture?

**Response:** Thank you for the detailed suggestion. We have added the essential information you suggested into the data resources to make it more complete.

**Revised (Lines 147-153 in the revised version):** Daily precipitation data were provided by the China Meteorological Administration (http://data.cma.cn/). In this study, only the stations (there are 824 basic weather stations in total) with complete data from 1980–2010 were selected. Climate Prediction Center (CPC) soil moisture (SM) data were provided by the National Oceanic Atmospheric Administration (NOAA)/Oceanic and Atmospheric Research (OAR)/Earth Systems Research Laboratory (ESRL) Physical Sciences Division (PSD), Boulder, Colorado, USA, from the Web site https://www.esrl.noaa.gov/psd/. The monthly data set consisted of a file containing monthly averaged soil moisture water height equivalents with spatial resolution of  $0.5^{\circ} \times 0.5^{\circ}$ . The data is model-calculated and not measured directly.

**Comments:** Section 2.4: Rainfall data is only available for 1980–2010, but the event FFE analysis is carried out for 1950–2015. What is done for the periods where rainfall data is missing?

**Response:** Thank you for the comments. In the discussion, we selected the period of 1980–2010, which has a more obvious temporal variation, as the study object to analyse the relationship between climate indicators and FFEs. As for your confusion, we added some reasons for the selection of 1980–2010 period at the beginning of the Section "5. Discussion".

**Revised (Lines 420-427 in the revised version):** The long-term evolution trends indicated that the number of FFEs in all six geomor-regions were relatively low before 1980. On one hand, this variation may be attributed to poor data acquisition methods and inadequate data records before 1980, which may have resulted in the lower occurrence observations in the historical period to some extent. On the other hand, some climate factors, e.g. precipitation, may have had an effect on the FFEs. With the changing climate, increasing frequency of extreme precipitation may have caused greater numbers of flash flood disasters in recent years. To reveal the potential factors of the FFE trends and better understand the peaks in the FFE time series, we discuss the potential relationships between climatic factors and FFEs in the following section.

**Comments:** Figure 1: It is not clear what Figure 1c shows. What is the meaning of the blue polygon? **Response:** Thank you for the question. Figure 1c shows the accumulation number of FFEs in each month during 1950–2015, which can clearly reflect the intra-annual difference of the FFEs occurrence. To better display the intra-annual distribution of FFEs, we changed the radar chart into bar chart in the revised version.

**Comments:** Methods: Different methods are used but only for the trend analysis with Sen's slope, the significance is calculated. I feel that it is absolutely necessary to provide significance statements also for the other analyses (reported in 4.1, 4.3, 4.4).

**Response:** Thank you for the insightful suggestion. Actually, we have done the significance test for MK test and monthly frequency. The significance test for MK test was detected by Z-statistics. When the Z-statistics with the absolute value larger than 1.28, 1.64 and 2.32, it represents the trend goes through the 90%, 95% and 99% significance test, respectively. The significance test for monthly frequency was obtained by the lower and upper bounds ( $L_L^N$  and  $L_U^N$ ), which indicated the confidence of 95%. Besides, referred to Merz et al. (Merz et al., 2016), we explored the time scale (T = 1a, 2a, 3a, 4a, 5a) and the start aggregation time of the time series at the 95% significant level, the results were as follows. The results indicated that the flash floods occurrence was not sensitive to the time scale and the start aggregation time of the time series in most area of this study.

**Revised (Lines 389-401 in the revised version):** To examine the inter-annual FFEs clustering, D was calculated based on the FFEs occurrence frequency data. D quantifies the deviation of annual occurrence rates from the expected occurrence rates. A D value larger than 0 was observed for all six geomor-regions across China, indicating that inter-annual clustering played a dominant role in FFE occurrence (Figure 9). Extensive significant clustering, a large D value of up to 4, can be detected for the SEM, SWM, and TP geomor-regions. The largest D value (D = 12) observed in the Wujiang River, which is located in the middle of the Yangtze River, means that on average FFEs occurred 12 times more often in FFE-rich years than would be expected from the Poisson distribution. The clustering strength diminished from eastern to western China. Nevertheless, six of the watersheds exhibited statistically significant results of a negative D, which indicated that the inter-annual occurrence of FFEs in these watersheds were under-dispersed. In this study, the characteristic under-dispersion of FFEs was mainly identified in western China, including in NWB, NCP, TP, and SWM, appeared to have a more regular temporal pattern between years.